# Maximum vertical height during wing flapping of laying hens captured with a depth camera

Tessa Grebey[1¤a], Valentina Bongiorno[1], Junjie Han[1¤b], Juan Steibel[2], Janice M. Siegford[1*]

1 Department of Animal Science, Michigan State University, East Lansing, Michigan, United States of America, 2 Department of Animal Science, Iowa State University, Ames, Iowa, United States of America

¤a Current address: East Lansing, Michigan, United States of America
¤b Current address: Bayer Crop Science, Chesterfield, MO
* siegford@msu.edu

## Abstract

Cage-free housing systems for laying hens, and their accompanying guidelines, legislation, and audits, are becoming more common around the world. Cage-free regulations often specify requirements for floor space and cage height, but the availability of three-dimensional space can vary depending on system configurations. Little research has looked at how much vertical space a hen occupies while flapping her wings, which is arguably her most space-intensive behavior. Therefore, the objective of this study was to use a depth sensing camera to measure the maximum vertical height hens reach when wing flapping without physical obstructions. Twenty-eight individually caged Hy-line W36 hens at 45 weeks of age were evaluated. A ceiling-mounted depth camera was centered above a test pen and calibrated prior to collecting data. During testing, one hen at a time was placed in the test pen and recorded flapping her wings. From depth footage, the minimum distance between pixels was obtained for each frame, and we computed the maximum vertical height reached by each hen. Results for vertical space used during a wing flapping event showed that hens reached a maximum height of 51.0 ± 4.7 cm. No physical measures taken from hens correlated with maximum height obtained from the depth camera (P > 0.05). Hens in this study were from a single strain, were old enough to have keel damage, and were cage-reared and housed, preventing us from generalizing the results too far. However, depth cameras provide a useful approach to measure how much space laying hens of varying strains, ages, and rearing/housing methods need to perform dynamic behaviors.

## Introduction

Worldwide, about 3 billion of the 7.5 billion total laying hens are kept in cage-free housing systems [1]. The percentage of hens in cage-free systems varies among regions, reaching up to 99% in some European countries, and housing a substantial share of the hen population in other countries including Egypt (50%); Australia (45%); the United States (40%); Colombia, Nigeria, and Ghana (30%); South Africa (14%); and Canada (13%) [1,2].

**Data availability statement:** All relevant data are within the manuscript and its Supporting Information data file (SM1 Data File).

**Funding:** This study was supported by the Michigan Alliance for Animal Agriculture award #AA19-041 (MAAA; East Lansing, MI) and by the National Institute of Food and Agriculture, United States Department of Agriculture (USDA), Hatch projects #1002990 and #1010765. Any opinions, findings, conclusions, or recommendations expressed in this publication are those of the authors and do not necessarily reflect the view of the USDA or MAAA.

**Competing interests:** The authors have declared that no competing interests exist.

In some cases, cage-free systems may be implemented because of economic necessity or because they suit production needs of a region; in others the push to end the use of caged systems is driven by those advocating for improved welfare for hens on commercial farms [3–6]. For example, in a survey 70% of U.S. consumers stated their support for a shift to cage-free eggs [7]. Pressure to house hens in cage-free systems for welfare reasons often results in standards, certifications, laws, and corporate commitments intended to support this objective. At the heart of many of these policies or laws are decreased stocking density (more space per hen) and provision of resources to allow hens to perform key behaviors important to welfare [3,8,9,10].

In the United States, there are no federal regulations covering laying hen housing systems used by the egg industry, and the transition to cage-free eggs in the U.S. is occurring in piecemeal fashion [4,11,12]. Various international corporations have made commitments to using cage-free eggs, and several states have implemented legislation to ban conventional cages in egg production. These corporate and state-based commitments often affect producers in other states or areas of the world. For example, laws passed in the states of California, Massachusetts, Michigan, and Rhode Island, stipulate that hens must be able to fully extend their limbs without touching the side of their enclosure, and eggs produced elsewhere but sold in these states must meet these criteria. Producers under contract to produce eggs for companies with cage-free commitments must also comply with these requirements, regardless of whether they are in a state with no cage-free legislation.

Adding to the complexity of the situation is the wide variation in types of cage-free housing systems for laying hens, which range from floor pen type systems with a single raised platform where nests, perches, feed, and water are located to multi-tiered structures that can contain more than four levels and be over 4 m tall in total. In the case of floor pens, ensuring floor area is adequate to allow dynamic behaviors like wing flapping is sufficient. However, in multi-tiered systems, there must also be consideration of the amount of vertical space between tiers or between tiers and ceilings. Legislation in some countries requires a minimum of 45 cm height between tiers [8,9,13]. Some welfare certifications specify at least 50 cm (e.g., Compassion in World Farming [14]) but many cage-free guidelines do not explicitly indicate how much height should be provided within tiers (e.g., United Egg Producers [10]). Further, the addition of structures, such as ramps to ease hens' transitions among levels or perches to facilitate roosting, can also reduce the amount of open space within levels—in addition to structures such as water and feedlines designed to meet hens' needs. Therefore, the structure of the cage-free system must also be explicitly considered when determining whether the space available facilitates behaviors of importance or how many hens should be stocked in the system.

While allowing hens more freedom of movement is laudable, the lack of more precise language raises concerns for practical implementation [15]. For example, do the housing systems need to allow enough space for only a single hen to extend her limbs at a time, or must all hens in a flock be able to do so simultaneously without interference from other hens? (The latter would require far more space.) The wings of laying hens clearly require much more space to extend than their legs, but little is known either about the wing dimensions of laying hens or when, why, and how often they extend them. A hen is likely fully extending her limbs not only when stretching, but also during wing flapping, which involves horizontal and vertical spreading of both wings in front of and around her body [16,17]. Wing flapping is performed by chickens as a component of dust bathing [18], during wing-assisted jumping or running [19,20], as part of agonistic conflict [21], during escape attempts [22], in anticipation of positive reward [23], in response to frustrated motivation [24], and on its own as a dynamic comfort behavior [16].

In 2023, the European Food Safety Authority (EFSA) Panel on Animal Health and Animal Welfare highlighted the importance of hens' freedom of movement in cage-free systems, providing a behavioral space model to calculate the birds' space allowance and stocking density based on specific behaviors [13]. Calculations were part of a broader approach to improving birds' welfare by considering animal-based indicators including the ability to perform behaviors without constraint, for which wing flapping was used as an example [13]. Two studies that previously examined the space required by hens to wing flap were used as the basis of the EFSA wing-flapping space calculations [20,25]. Both research groups examined the horizontal area occupied by laying hens while flapping, concluding that between 1693.0 ± 136.0 cm² [20] and 3344.5 ± 92.3 cm² [25] of floor area was needed, with brown feathered hens requiring less horizontal space than white feathered hens [25]. However, only Mench and Blatchford [20] have published on the amount of vertical space used by laying hens during wing flapping, reporting that Hy-Line W36 hens used 49.5 ± 1.8 cm of vertical space while flapping their wings when jumping down from a perch. However, the angle at which wings are raised, and the subsequent maximum height, varies depending on the function of the flap during locomotion (e.g., [19]). Following this reasoning, wing flaps performed when a hen is stationary, such as for the purpose of stretching, might also occupy different amounts of vertical space than wing flaps during locomotion. Therefore, it would be important to assess and incorporate various types of wing flapping into space calculations, guidelines, and equipment design.

Technology allows new insights into the study of animal behavior, including behavior of poultry [26,27]. Modern imaging technology has the capacity to capture high speed motions in three dimensions and can be used to assess the amount of space occupied by a hen while wing flapping. Given that more hens are being housed cage free across the world, it is important to know how much space a hen occupies while wing flapping to assist both those developing standards or enforcing legislation, as well as equipment manufacturers, builders, and producers as they design and implement aviaries. Use of depth or high-speed cameras could enable researchers to accurately estimate space used by hens of various strains to perform a range of dynamic behaviors. Coupled with information about how often hens do such behaviors and whether there are elements of behavioral synchrony or social facilitation or if there is a circadian rhythm motivating performance of the behavior, space requirements for the behavior could be used to design housing systems and provide stocking density recommendations. However, as it may not be possible to use specialized cameras to study all possible strains of hens or hens of various ages, understanding the relationship between anatomical measures from the hen (such as wing length or body weight) and images could enable, simpler and less costly estimates of space to be obtained. For example, allometric proportions are often observed between wing measurements and body weight in birds [28], and in free-range chickens, physical measurements can predict birds' body weight [29]. However, laying hens are simultaneously impacted by genetic selection for continuous egg production and uniform body weight while flight has not been under selection pressure. Furthermore, hens may have developed different adaptation strategies to cope with environmental challenges in artificial environments. For example, previous studies have demonstrated differences in the wing area and wing load between laying hens and jungle fowl, that reflect the greater flying abilities of the wild ancestral species [30,31]. Moreover, laying hens on commercial farms may preferentially use terrestrial pathways to navigate through cage-free systems when possible, rather than using vertical movements that necessitate wing use [32], and this may be more pronounced in brown compared to white hens [33]. For these reasons, a correlation between birds' body measurements may be present but not obvious, contrasting with what is observed in wild birds [34].

The primary objective of this study was to describe the maximum vertical height needed for laying hens to wing flap using a depth camera. Secondarily, we hypothesized that despite genetic selection, physical measures of hens' body weight and wing dimensions would correlate to the maximum wing-flapping height observed. Findings from this study may provide some idea of three-dimensional space requirements for wing flapping and provide guidance for other technology-based methods to address other space use questions in laying hens.

## Materials and methods

### Ethical approval

Prior to beginning research, study protocols were evaluated and approved by the Michigan State University Institutional Animal Care and Use Committee (PROTO202100131).

### Animals and housing

A flock of Hy-Line W36 pullets were originally reared together in pullet cages at the Michigan State University Poultry Teaching and Research Center in East Lansing, MI. Subsequently, these hens became part of the fertile egg flock, used to generate fertile eggs sold to biomedical researchers. To ensure insemination of hens and to facilitate egg collection, hens were individually housed in cages (30.5 x 46.5 x 42.0 cm). The floor in each cage sloped to allow eggs to be easily collected, resulting in slightly more vertical space (~3 cm) near the front of the cages compared to the back. For the purpose of this project, twenty-eight of these hens were used at 45 weeks of age. No birds were anesthetized or killed for this study.

### Physical measures from hens

Two persons trained in humane handling techniques collected physical measures of body weights and wing dimensions from each hen. Hens were removed one by one from their home cages and held upright by one person, so that their right side was pressed against the person holding them and their left wing could be handled freely by the person taking the measurement. Physical measurements were taken from the left wing of each hen. First, a 1 m long wooden ruler with 0.5 cm gradations was held gently over the outer surface of each hen's folded wing to assess the length from carpal joint to the distal tip of the longest primary feather (Fig 1A). The wing was then fully extended horizontally from the hen, and the ruler was held underneath the wing with the ventral surface of the wing resting on the ruler and the wing gently flattened. A measurement was taken from the point where the wing joins the body to the distal tip of the phalanges to record the length of the skeletal structure (Fig 1B). This was done by the person conducting the measure crouching down to enable them to look up from underneath the wing so that ceiling lights shone through primary feathers to indicate where the bone ended at the distal tip of the longest phalange. With the ruler still held in this position and the left wing still fully extended, a measurement was taken of the distance from the body to the distal tip of the longest primary feather (Fig 1B). Each hen was weighed by placing her gently in a bucket on a digital scale before she was returned to her home cage.

### Depth camera and experimental set up

An Intel RealSense Depth Camera D435i (Intel Corporation, Santa Clara, CA) was used to film hens wing flapping. This depth camera (90 x 25 x 25 mm) was mounted to the ceiling in the same barn as hens' home cages (Fig 2A). A black plywood board (121 x 121 cm) was affixed to the floor directly underneath the camera; this board provided contrast against the white-feathered hens. The face of the camera was 250 cm from the surface of the plywood. The

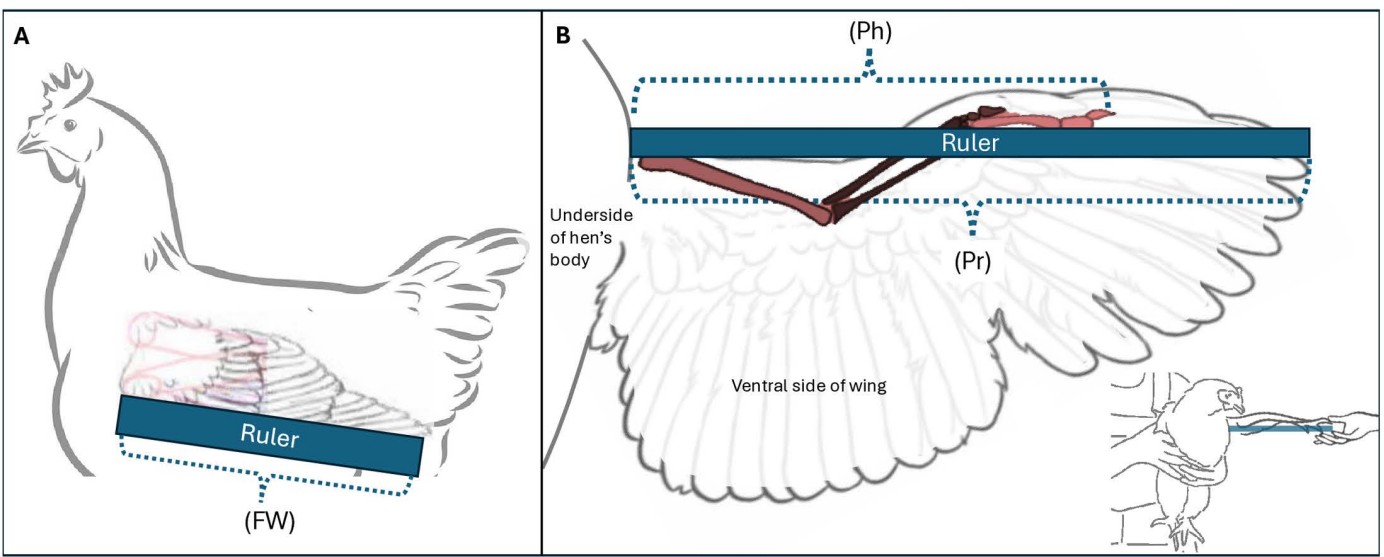

**Fig 1. Schematic illustrating where physical measures were taken of hens' wings. A.** The length of the folded wing (FW) was obtained by placing the ruler over the wing and measuring from the carpal joint to the tip of the longest primary flight feather. **B.** Measures of manually extended wings were taken by placing a long wooden ruler against the ventral (underside) of the bird's wing. First, the wing was gently extended at a right angle from the body. Then, the end of the ruler was placed against the hen's body under the wing, where the wing joined the body. Distances were measured from the body to distal end of the phalanges (Ph), marking the extent of skeletal structure of the wing, and from the body to the end of the longest primary flight feather (Pr), marking the extent of the feathers of the wing.

depth sensor of the Intel camera was set to be 90 frames per second with a resolution of 640 x 480, and the RGB camera was set to be 30 FPS with a resolution of 1280 x 720. Frames from the depth sensor were downsampled to allow them to be paired with RGB frames, while RGB frames were cropped and mapped to depth frame pixels. Lights were placed around the testing area to prevent shadows in the camera's field of view. Prior to collecting any data, the depth camera was calibrated to the flat surface of the plywood board using objects of known size (such as a 5-gallon bucket) to ensure the camera was correctly reporting the depth. A foldable wire pen (122 x 122 x 122 cm) was secured around the sides of the plywood after calibrating the camera to keep hens in the camera's field of view. The pen was open on top, giving the depth camera an unobstructed top-down view of each hen during testing, and the sides were tall enough to prevent hens from flying out. An RGB video camera (Canon VIXIA HF M41 A, Tokyo, Japan) was positioned on a tripod at hen height, facing the test pen to film each hen from a side angle rather than top down as she flapped her wings (Fig 2B). Video from this camera was recorded at 30 frames per second and was used to help pinpoint when wings were extended upward during wing flapping.

We encouraged wing flapping by placing hens individually into small, hard-sided pet carriers (45.5 x 29.2 x 30.5 cm) to restrict their space prior to moving them into the pen for testing. These carriers were smaller than hens' home cages, and we expected that rebound effect would lead to hens stretching and flapping their wings promptly once they had room to do so in the test pen. Hens spent between 10-60 minutes in the carrier, as described in more detail below in the test procedure section.

## Wing flapping experimental procedure

Wing flapping can occur in several forms. For our study, we recorded only stationary wing flapping events when a hen was standing upright in a stationary position with her feet on the

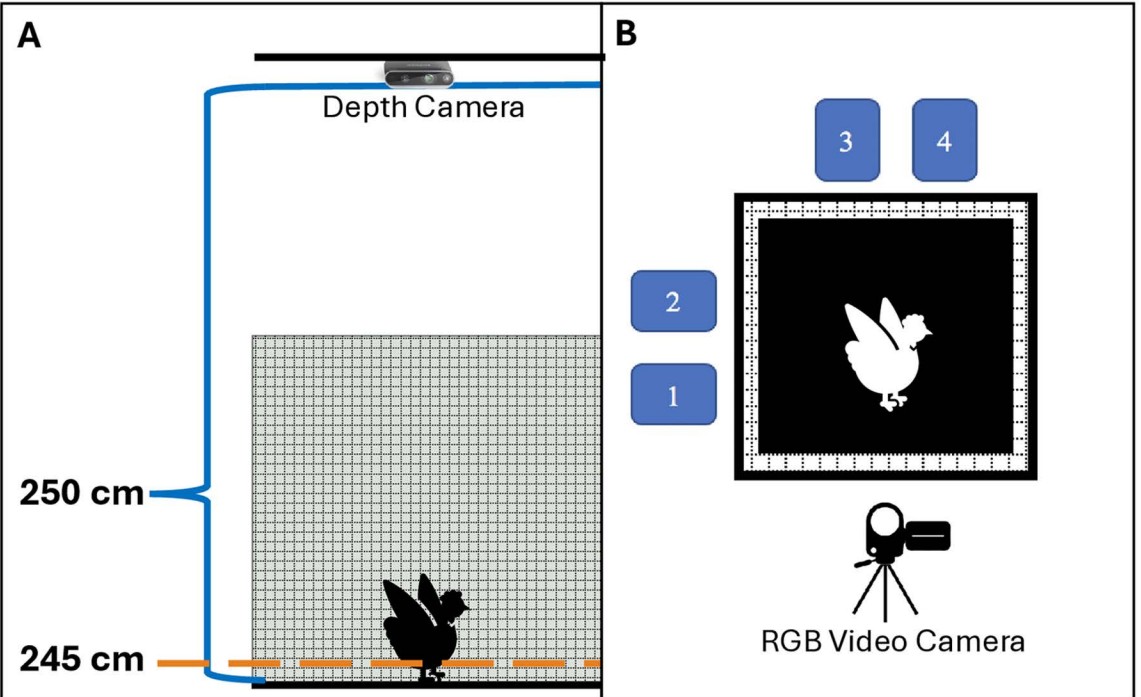

**Fig 2. Schematics of the experimental set up. A.** Side view of test pen with a hen. The depth camera was centered directly above the test pen, 250 cm above the surface of the black-painted plywood floor. The orange line represents the threshold applied for depth video analyses, which was 5 cm above the floor of the pen. **B.** Top-down view of the test pen. The blue rectangles labelled 1-4 represent the small carriers that hens were placed in prior to testing; these carriers faced the test pen allowing hens to see other hens during holding and testing. An RGB video camera on a tripod recorded the hen from the side to confirm wing flapping and likely time of maximum vertical extension.

ground while extending both of her wings away from her body at least one time (adapted from Riddle et al. [25]). If the hen flapped her wings repeatedly without pause, we recorded all of the wing flapping bout. If the hen took only a step or two for balance during the flapping event but did not appear as though her wings were raising her or propelling her forward, the flap was recorded. Thus, the stationary wing flap is different from wing-assisted locomotion events when a hen uses her wings to help her move through her environment [35].

Two people trained in humane handling of poultry handled hens and conducted the testing procedure. Hens were habituated to the procedure starting one month before data collection and each hen was habituated twice. Habituation procedures mimicked the testing procedure and included all steps from removal of individual hens from their home cages to placement into carriers and concluded with 10 minutes in the test pen before being returned to the carrier and then her home cage. All 28 hens were tested on the same day for consistency in lighting and depth calibration. Hens were removed from home cages in groups of four and were placed individually into one of the four small carriers. These carriers were situated around the test pen so that each group of four hens had visual access to one another (Fig 2B).

Hens remained in the carriers for at least 10 minutes to allow study personnel to record hens' numbers and to restrict hens to encourage quick wing flapping. After spatial restriction, one person removed the first hen in the group from her carrier and placed her into center of the test pen. As soon as the handler stepped away from the test pen (to avoid shadows in the image), the second person simultaneously began the video recording by the depth and video

cameras. Each hen remained in the test pen until she flapped her wings, or 10 minutes elapsed. If a hen did not flap her wings within 10 minutes, she was returned to her carrier and video files from that attempt deleted. Once other hens in that group were tested, the non-flapping hen was placed back in the test pen for a second test attempt. When a hen successfully flapped her wings within the allotted time, the depth camera and video camera were stopped, and both video files were saved and labelled with date and hen ID. If the hen flapped her wings repeatedly without pause, the entire wing flapping bout was recorded. The hen was then returned to her carrier. Once all 4 hens in a group were recorded flapping their wings in the test arena, they were returned to their home cages and another group of 4 hens were removed for the test procedure. This was repeated until all 28 hens were recorded flapping their wings. No hens were kept away from their home cages (i.e., feed and water) for more than 1 hour. Between 3-7 hens were tested per hour (depending on how quickly hens flapped and whether more than one attempt was needed for a hen).

## Analyzing depth videos

Each of the 28 videos were recorded using the Intel® RealSense™ Viewer software (Intel Corporation, Santa Clara, California, USA) and saved in rosbag format. This type of file (rosbag) has a large file size compared to a typical .mp4 formatted file, but it contains all required data and meta data to re-construct an RGB image as well as a depth map. MATLAB (MathWorks, Natick, Massachusetts, USA) was utilized to read the rosbag files and to perform all subsequent processing described below. First, RBG and depth streams were aligned and frames from each video file were cropped to only feature the hen standing on the black plywood while flapping. We then performed a post-hoc analysis of depth streams and applied a distance-based threshold so that each depth video contained only those instances (pixels) when the depth values of the pixels fell in the threshold range. In this case, as the depth camera was 250 cm above the ground, the threshold was set to 245 cm. This threshold generated a binary mask to segment out pixels corresponding to the body of the hen to be included for analysis (i.e., hens' bodies reached a high enough vertical point to be captured by the depth camera and was separated from the more distant black plywood). The binary mask (yellow, Fig 3A) was then applied to the aligned RGB frame for visual verification that no 'hen pixels' had been missed (Fig 3A). Finally, all overlapped depth frames, overlapped RGB frames, and the masked RGB frames were concatenated and saved into a new .mp4 video. In the masked RGB, a red dot was placed to note the pixel that represented the highest point in each frame. In this way, when a video of a hen wing flapping is played, the red dot moves with the hen to indicate which part of her body was at the highest vertical point in each frame while wing flapping (Fig 3A). The vertical height of each pixel was calculated from the floor (i.e., by subtracting it from 250 cm), and the pixel with the maximum height for each frame was extracted and visualized (red dot, Fig 3A). Images from the RGB camera located beside the pen were used to confirm vertical wing extension (Fig 3B). As, our aim was to detect the maximum height during the wing-flapping bout, we did an across-frame maximum height extraction.

## Statistical analyses

The weight, length of folded wing (carpus to primary tip), length of extended wing to distal tip of longest primary, and length of extended wing to distal tip of phalange were recorded from 28 hens and analyzed with by means of IBM SPSS Statistics (International Business Machines Corporation, Armonk, New York, USA). An outlier assessment was carried out, and one measurement within the extended wing (to primary tip) data was detected as outlier and removed. Data were tested for normality, and as data were not normal, the non-parametric

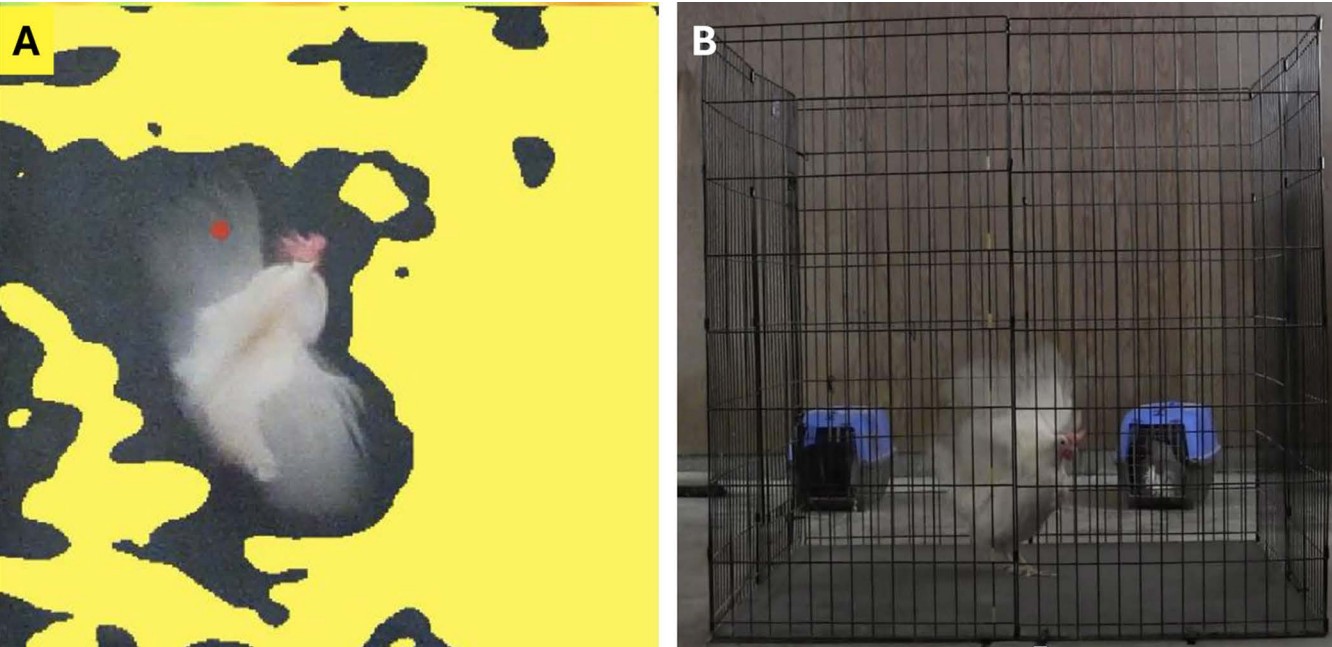

**Fig 3. Examples of images used to measure vertical height reached by wing flapping hens while in the test pen. A.** Image taken from the depth video file. The binary mask removed pixels (yellow) which were below the 245 cm threshold and were not included in depth analysis. The red dot indicates the part of the wing flapping hen's body that was the highest vertical point in this frame. **B.** Image from the RGB camera placed beside the test pen corresponding to the example images captured by the depth camera.

Shapiro-Wilk test was selected due to the reduced sample size. Furthermore, a linearity test was performed and scatter plots of the variables and their residuals generated. Descriptive statistics were subsequently generated by the software. The variables were then tested for linearity, and, as the assumptions for a Pearson correlation test were not satisfied, a Spearman correlation test was performed on paired physical measures. As no significant results were found, a correction for multiple comparisons to reduce Type I error was not carried out.

## Results

Once placed in the test pen, hens took between 35 seconds to over 7 minutes to successfully flap their wings. Two of the 28 hens did not flap their wings within the 10-minute time limit, but upon retesting these 2 hens did flap their wings. Wing-flapping hens reached an average maximum height of 51.0 ± 4.7 cm (Fig 4A.).

Averages of the physical measurements and maximum vertical height from the 28 hens are presented in Fig 4 (A-E). No significant correlations were discovered between the maximum height obtained from the depth camera and the physical measurements of hens (Table 1, S2_Fig). When considering the possible combinations between the various physical measures, a correlation trend was found between the length of a hen's folded wing and the length of her extended wing extended to the tip of the longest primary feather (r = 0.36, p = 0.06).

## Discussion

The hens examined in the present study used an average of 51.0 ± 4.7 cm of vertical space while flapping their wings. Our results are similar to those reported by Mench and Blatchford [20], who found a maximum wing flapping height of 49.5 ± 1.8 cm in their Hy-Line hens. Together these findings suggest that systems offering < 55 cm of usable, vertical space between

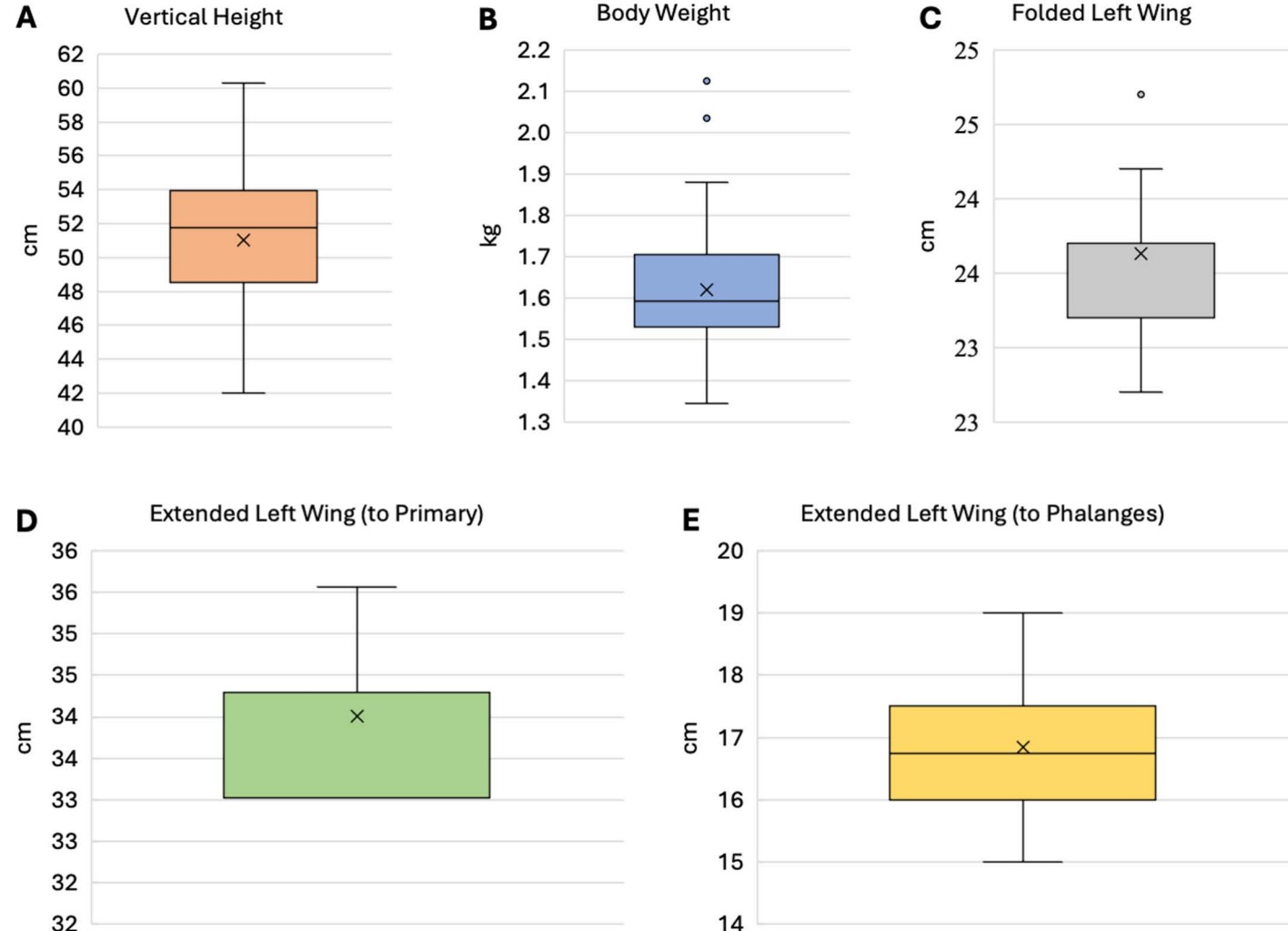

**Fig 4. Descriptive statistics for vertical height reached by 28 Hy-Line W36 hens while wing flapping along with physical measures of weight and wing dimensions. A.** Maximum vertical height reached by the hens while wing flapping (cm) as recorded with a depth camera. **B.** Body weight (kg) of hens in the study. **C.** Length of the folded wing as measured from carpal joint to distal tip of the longest primary feather (cm). **D.** Length of the extended wing measured from where the wing joins the body to the distal tip of the longest primary feather (cm). **E.** Length of the extended wing measured from where the wing joins the body to the distal tip of the phalange (cm). All wing measurements were taken from the hen's left wing. Bold horizontal line = median. X = mean. Vertical lines extending from the box plots = standard deviations.

levels or between the lowest tier and the floor likely do not allow hens, at least of this strain, to wing flap without touching the ceiling in those areas.

It should be noted that our hens and those studied by Mench and Blatchford [20] were older birds (45 and ~78 weeks of age, respectively) and had been reared and housed in cages. It is therefore probable these hens had limited or impaired mobility due to prolonged confinement and age. Moreover, it is possible that hens from our study were never afforded enough three-dimensional space to lift their wings other than the brief time they spent in the test pen, as they were housed in cages of 30.5 x 46.5 x 42.0 cm. Therefore, the manner in which they flapped may not be representative of hens with more muscle development and practice wing flapping. In cage-free systems, hens have more opportunity to use and develop their muscles [36], including breast muscles that assist with wing flapping [37]. However, previous studies showed that birds may adapt their strategy for transitioning among levels in the system rather

**Table 1. Spearman rank correlations between physical measurements taken from hens.**

| Physical Measures Correlated | r | P-value |
|---|---|---|
| Weight x Folded wing (carpus to primary tip) | 0.120 | 0.542 |
| Weight x Extended wing (to primary tip) | 0.176 | 0.379 |
| Weight x Extended wing (to phalanges) | 0.052 | 0.792 |
| Folded x Extended wing (to primary tip) | 0.357 | 0.062 |
| Folded x Extended wing (to phalanges) | 0.044 | 0.824 |
| Extended wing (to primary tip) x Extended wing (to phalanges) | 0.270 | 0.173 |
| Height x Weight | 0.148 | 0.450 |
| Height x Folded wing (carpus to primary tip) | -0.001 | 0.998 |
| Height x Extended wing (to primary tip) | 0.169 | 0.389 |
| Height x Extended wing (to phalanges) | 0.259 | 0.184 |

[a]Maximum height during wing flap measured using depth camera.

than altering their muscle development [32]. Future studies utilizing depth cameras to evaluate dynamic behaviors could examine younger hens and hens of various other strains, as well as birds reared and housed in cage-free systems they would have better muscle development and experience with wing flapping. All of these factors are likely to influence how much space a hen uses when wing flapping and would provide a more nuanced picture.

No significant relationships were found among the various measures of hens' wings or between hens' wing measures and their body weights. The strongest correlation (r = 0.357) was observed between the measurement of the folded wing and that of the extended wing (through the distal tip of the longest primary feather) though this was only a trend toward significance. In this study we compared the various measures to one another hypothesizing that all would be related, by virtue of the harmonious body which characterizes laying hens despite being commercial strains. We acknowledge that performing multiple comparisons increased the likelihood of a finding a correlation where none existed (a Type I error), but as no significant correlations were detected, making corrections for multiple comparisons would not have impacted our results. A study based on a larger sample size would be needed to confirm or discard this trend toward significance. The lack of proportional relationship was surprising when comparing hens' weight to wing measures, as wings are supportive structures in birds' locomotion [38]. Pennycuick, [28] reported that wing area and wingspan can vary broadly and independently from each other in several bird species, but that allometric proportions are typically observed between such measurements and birds' body weight. Similarly, Tyasi and colleagues [39] found significant correlations between physical measures and body weight in free-range chickens. However, we used a relatively small sample of hens with little variation in body weight and greater variation in wing measures and may not have had the power to detect a relationship. Additionally, laying hens are *Galliformes*, which are ground birds whose wing use is dedicated to bipedal movement assistance and limited short flights [38]. For this reason, in comparison with strong fliers, hens may not require a proportional variation in wing dimension in relation to their weight. As explained above, the absence of correlation between maximum vertical height while wing flapping and the birds' weight and physical measures could also be explained by the limited vertical space available to the hens in their housing. The home-cage dimensions in the current study were too small to allow hens to flap their wings, certainly not with full wing extension. Such a factor may have altered their wing flapping performance in ways that do not relate to underlying body weight and physical measures of their wings.

When a hen flaps her wings, she not only requires vertical space but also adequate free horizontal space, often referred to as usable floor space when discussing space requirements of cage-free hens, which is critical to understand in determining stocking densities in housing systems [13,40,41]. Previous studies on horizontal wing flapping space use have reported hens of this same strain (Hy-Line W36) using 3344.5 ± 92.3 cm$^2$ [25] and 1693.0 ± 136.0 cm$^2$ [20] of horizonal space. The variation among previous results may be explained by the rearing and housing of the hens used and conditions under which data were collected. Riddle and colleagues [25] collected measurements from hens who had been cage-free reared and housed while they were performing stationary wing flaps on the floor in an open litter area of their home aviary. Mench and Blatchford [20] collected data from hens that were cage-reared and housed while jumping from a perch in a test chamber. Further, in Riddle et al. [25], the authors deliberately focused on the phase of the flap occupying the most horizontal area. As different wing-based behaviors require varying amounts of three-dimensional space, it is possible that hens use less room when flapping their wings for jumping purposes [20] as opposed to when they are standing stationery and flapping, when they may be more likely to fully extend their wings as part of a stretch or display [16,21]. Our current findings on maximum vertical height while wing flapping by Hy-Line W36 can be used in conjunction with the maximum horizontal area used by wing flapping of hens of the same strain reported by Riddle et al [25]. Together these measurements can provide an estimate of the 3D space occupied by a wing-flapping hen to better evaluate space needed to perform this dynamic behavior.

The amount of space a hen occupies when performing dynamic behavior also varies depending on genetic strain [25]. There are also notable genetic differences in hens' skeletal and musculature structure that may influence how they perform specific behaviors, like wing flapping. White-feathered hens tend to have heavier breast muscles and larger keel bones, whereas brown-feathered hens have heavier leg muscles [42,43]. More research must be done to assess space requirements across multiple strains of laying hens before comprehensive guidelines can be provided, and it is possible that a range of height guidelines are necessary for hens of different strains.

Finally, 26 of the 28 hens tested wing flapped between 35 seconds and 7 minutes. The two birds who did not wing flap during the initial 10-minute testing period did so when placed in the test arena a second time. The hens in this study were reared in cages, with no opportunities to exercise their wings over time. Nevertheless, in a relatively short amount of time all the birds wing flapped. This finding implies innate motivation for such a behavior in the hens, and the need to extend their wings soon after being placed in the test arena was probably consequential to cage housing. Similarly, space availability is a relevant factor shown to affect the behavioral performance of hens in previous studies [40]. Anecdotally, the tested hens appeared inexperienced while wing flapping; showing uncoordinated movements like slipping, sliding, and tilting that are usually observed in young pullets (personal observation). This observation reinforces the hypothesis that long-lasting effects of cage rearing systems may impair the birds' physical ability to wing flap.

## Additional considerations

It remains unclear if spacing requirements in cage-free legislation should require that a hen (or all hens in a group) be able to flap her wings at all times. Hens may not need or want to flap their wings at certain times of day because they are performing other behaviors that cannot occur concurrently, such as eating or laying eggs. Additionally, laying hens do not distribute themselves evenly throughout a given area but often cluster to access resources or to maintain proximity to each other [44]. When hens congregate within a housing system, they effectively increase the stocking density of that immediate area, which can reduce their

freedom of movement in that space [41,45]. Crowding may limit a hens' actual ability to wing flap at a given time in a particular place, or she may not think she is able to flap due to perceiving the proximity of her flock mates [46]. The function, motivation, and causation of wing flapping should be further examined so that cage-free initiatives can clearly drive practical improvements to cage-free husbandry and housing that genuinely benefit hens.

## Conclusion

Multiple conclusions can be drawn from the present research. Overall, this study reinforces the concept that a minimum of 56 cm of usable, vertical space is required for Hy-Line W36 hens to wing flap freely within the various areas and levels of cage-free systems, to avoid touching the ceiling of those areas. The space occupied by the hens while wing flapping provides an interesting perspective on current behavioral opportunities (or lack thereof) within cage-free aviary systems, where heights between levels may be less than what a hen physically needs to wing flap or perform other similarly spatially intense behaviors. Moreover, further studies need to assess the hens' perception of the space provided beneath and within the tiers, which despite appearing to be physically sufficient to transit or stand may not be perceived as comfortable or as enough space to allow wing flapping or static stretching. Thus, the very cage-free systems intended to improve laying hens' welfare might not be working as intended. Additional research on hens of different genetics and ages is required, encompassing strains which likely display greater muscle strength and flapping abilities (e.g., local breeds and genotypes less selected for commercial egg laying), to complete our understanding of the space needed for vertical wing extensions. Furthermore, depth camera data and physical measures collected from the hens were mostly uncorrelated in the present study. If this is the case, it hampers the use of easier-to-collect physical measures from hens of different ages and strains to predict space needed for wing flapping. However, data from a larger, more diverse sample of hens might find patterns not detected here. Finally, depth cameras, such as the one used here, appear to be useful as a supporting technology for behavioral studies of hens. Their use allowed us to successfully capture wing flapping events and generate measurements on the space required for hens to wing flap.

## Supporting Information

**S1 Data. Wing flapping depth data and physical hen measures.** Raw data analyzed and presented in the manuscript are available in the two worksheets of this supplemental data file.
(XLSX)

**S2 Fig.** Plots showing Spearman rank correlations between physical measurements taken from hens.
(DOCX)

## Acknowledgments

The authors thank the Michigan State University Poultry Teaching and Research Center manager Angelo Napolitano for his assistance with study set up and research assistant Elizabeth Gregas for her help in data collection.

## Author contributions

**Conceptualization:** Tessa Grebey, Juan Steibel, Janice M. Siegford.

**Data curation:** Tessa Grebey, Valentina Bongiorno, Junjie Han, Janice M. Siegford.

**Formal analysis:** Tessa Grebey, Valentina Bongiorno, Juan Steibel.

**Funding acquisition:** Janice M. Siegford.

**Investigation:** Tessa Grebey, Junjie Han, Janice M. Siegford.

**Methodology:** Tessa Grebey, Junjie Han, Juan Steibel, Janice M. Siegford.

**Project administration:** Janice M. Siegford.

**Resources:** Juan Steibel, Janice M. Siegford.

**Software:** Juan Steibel.

**Supervision:** Juan Steibel, Janice M. Siegford.

**Validation:** Tessa Grebey, Valentina Bongiorno, Junjie Han.

**Visualization:** Tessa Grebey, Valentina Bongiorno, Junjie Han, Juan Steibel, Janice M. Siegford.

**Writing – original draft:** Tessa Grebey, Valentina Bongiorno, Janice M. Siegford.

**Writing – review & editing:** Tessa Grebey, Valentina Bongiorno, Junjie Han, Juan Steibel, Janice M. Siegford.

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
