## [Decision Letter · Decision Letter 0]

6 Dec 2024

PONE-D-24-45495Maximum vertical height during wing flapping of laying hens captured with a depth cameraPLOS ONE

Dear Dr. Siegford,

Thank you for submitting your manuscript to PLOS ONE. After careful consideration, we feel that it has merit but does not fully meet PLOS ONE’s publication criteria as it currently stands. Therefore, we invite you to submit a revised version of the manuscript that addresses the points raised during the review process.

**Based on the reviewers' comments, major revisions are required to address key concerns. These include clarifying methodological details (e.g., wing flapping definitions, measurement methods, and statistical analyses), providing stronger justification for the study’s objectives, and addressing study limitations, such as focusing on a single strain and individual housing. Additionally, consider analyzing horizontal space or clearly justify its exclusion. We look forward to receiving your revised manuscript addressing these points.**

We look forward to receiving your revised manuscript.

Kind regards,

Arda Yildirim, Ph.D.

Academic Editor

PLOS ONE

**Journal Requirements:**

2. To comply with PLOS ONE submissions requirements, in your Methods section, please provide additional information regarding the experiments involving animals and ensure you have included details on (a) methods of sacrifice, (b) methods of anesthesia and/or analgesia, and (c) efforts to alleviate suffering.

This study was supported by the Michigan Alliance for Animal Agriculture award #AA19-041 (MAAA; East Lansing, MI) and by the National Institute of Food and Agriculture, United States Department of Agriculture (USDA), Hatch projects #1002990 and #1010765. Any opinions, findings, conclusions, or recommendations expressed in this publication are those of the authors and do not necessarily reflect the view of the USDA or MAAA.

**Additional Editor Comments:**

Dear Janice Siegfor and co-authors,

The reviewers have found your study to be relevant and impactful, addressing an important issue in vertical space requirements for wing flapping in laying hens. However, several key areas need clarification and improvement before the manuscript can be considered for publication. These include providing additional methodological details (e.g., definitions and measurements of wing flapping and statistical analyses), addressing study limitations (e.g., focusing on a single strain and individual housing), and expanding on the rationale and discussion of your objectives. Furthermore, reviewers noted the need to either analyze horizontal space or justify its exclusion. Enhancing the clarity of data presentation (e.g., through figures or supplementary materials) and ensuring consistency in terminology and grammar throughout the manuscript are also critical. I encourage you to incorporate these suggestions into a revised version of your manuscript. Thanks. Regards, Arda Yıldırım

Reviewers' comments:

Reviewer's Responses to Questions

**Comments to the Author**

1. Is the manuscript technically sound, and do the data support the conclusions?

Reviewer #1: Yes

Reviewer #2: Partly

Reviewer #3: Yes

Reviewer #4: Partly

Reviewer #5: Yes

2. Has the statistical analysis been performed appropriately and rigorously? 

Reviewer #1: Yes

Reviewer #2: N/A

Reviewer #3: Yes

Reviewer #4: I Don't Know

Reviewer #5: Yes

3. Have the authors made all data underlying the findings in their manuscript fully available?

Reviewer #1: Yes

Reviewer #2: Yes

Reviewer #3: Yes

Reviewer #4: Yes

Reviewer #5: Yes

4. Is the manuscript presented in an intelligible fashion and written in standard English?

Reviewer #1: Yes

Reviewer #2: Yes

Reviewer #3: Yes

Reviewer #4: Yes

Reviewer #5: Yes

5. Review Comments to the Author

**Reviewer #1: ** Comments to the manuscript PONE-D-24-45495

The "Introduction" chapter is well written and contains enough information to support the validity of the study. However, I miss a clear hypothesis. The authors constructed the purpose of the study and in principle one can infer from the text what their hypothesis is, but it is not directly stated. In my opinion, the authors want to confirm or rule out that the use of depth cameras will be a better tool for measuring the vertical height of the aviary needed for wing flapping than absolute measurements of the body dimensions of laying hens.

L95: Does this mean it was a parental (reproductive) flock?

L95: Why was it decided to use Hy-Line hens despite the fact that the Authors indicated that similar studies had already been conducted for this genetic group (L69) and the influence of the hens' genotype on the aviary/cage space requirement had been confirmed (L67)? Wouldn't it be better to choose another laying hens' genotype that had not been studied in this respect?

L97-98: I am not sure whether keeping hens individually in cages for the experiment was the right choice. Yes, it will give an answer about how much space a single hen needs, but as the Authors themselves mentioned in the introduction, it is important to indicate whether all the hens in the cage will be able to flap their wings simultaneously (L47-49). Considering that in commercial keeping of laying hens they are never kept individually, a group assessment seems more important!

L104-117: Why were only the wing dimensions measured? In my opinion, the entire body of the hen should be measured, i.e. from the foot to the end of the longest flight feather (when the hen is standing). The wing span (both) provides information on the width of the cage required, but for assessing the height required, the span of the hen's body from the foot to the end of the wing stretched upwards is important. Furthermore, I do not know whether body weight provides significant data in this case because it may not be related to body dimensions, e.g. hens with more fat and those with an egg in the shell gland will be heavier than others, even though their body dimensions may be smaller.

**Reviewer #2:**  General

This paper addresses an interesting issue in space requirements, particularly vertical space, for lying hens while flapping their wings. There are two objectives to this study: 1) to assess the vertical space needed for hens to wing flap, and 2) to correlate BW and wing dimensions to maximum wing flapping height. For this purpose, the authors observed 28 Hy-Line W36 hens at 45 weeks of age, measured wing dimensions, and recorded them performing wing flapping behaviour with a depth camera.

The study is perhaps a little bit premature as it focussed on one strain and age; it would have been interesting to expand and e.g., assess vertical space over multiple strains and time points. That said, the study is well-described and a necessary first step. The authors also acknowledge the limitations and take care not to overstate their results. The paper is generally easy to read and follow, though I do have some points that I think should be clarified.

I have provided some of my main questions and concerns here, while more minor comments are described later.

1. Figure 1 and 2 do a really nice job of visualizing the experimental set up for the reader, but some images/schematics (e.g., a photograph with overlaid measurement indicators) to really show what is being measured with the ruler (and camera) would be a valuable addition. Reading some of the descriptions e.g., ‘length from carpal joint to the distal tip of the longest primary feather’ (L109), ‘where the muscle and bone ended at the distal tip of the longest phalange’ (L113), ‘measurements to the distal tip of the longest primary feather’ (L115), I lost oversight on what for example the differences was between what is being described in L113 and L115. Currently, it was not fully clear me e.g., where do you start/end the measurements, did you measure to a specific primary feather, a straight line from A to B or some other angle/method e.g., 90 degree from body centre etc.? It would help put the described measures into perspective and also make it easier to link/interpret the results on the correlations in Table 2.

2. I missed the rationale for the second objective (correlations between BW/wing dimensions and vertical height) which I think should be woven into the introduction / paper to help strengthen the story. Currently, it felt a little like it was there for the sake of having done some analyses. Some explanation on why you wanted to look at e.g., correlations between BW and wing dimensions, would be useful also to justify the multiple correlations evaluated (and the risk of finding correlations by chance). The ones between maximum height and the other variables are more intuitive to me but would also benefit if the authors could make this more clear and explicit throughout the paper.

3. The authors chose to focus on vertical space but I think it would have been really useful if the authors could also elaborate on the horizontal space. It might add useful insights if hens that need more vertical space also need more horizontal space as often assumed, but I have also heard of discussion where taller animals are thought to actually be slimmer/less dense and so taking up less space horizontally. If possible, I think this would really add value to the paper to also analyse horizontal space. If not possible, then I think the authors should carefully make their case for why it was not included.

Introduction:

There is a strong focus on the hens in cage-free housing throughout the introduction. However, would the space required not also apply to birds in caged housing system even if they have less opportunities to perform the behaviour?

L41: should this not be export?

L46-49: I think this is a fair question to raise, however its placing here made me think it would be addressed in the study. Perhaps nuance it a bit for some expectation management, or leave only a short mention the discussion?

L50: Wouldn’t the wings also require much more space in the horizontal place? You refer to this in L53

L51: a research group at the University of Guelph has done some studies in recent years on wing use/disuse in laying hens and its biological implications. In some they also reported on wing length and wing beat frequency, as well as musculoskeletal development. That may be useful to support the authors statements here, but more importantly give a bit more depth and up to date literature to the discussion especially around L243-247; L256-266 considering the angle taken there aligns well with these topics. See e.g., https://doi.org/10.1098/rsos.220809; https://doi.org/10.1098/rsos.230817; https://doi.org/10.1098/rsos.210196. I will leave it up to the authors to decide, but I wanted to point it out in case the authors were unaware.

L62: I do not fully understand how the ‘while reducing plumage damage’ comes into play?

L72-73: Could you elaborate on this? I am not sure how it leads to the suggestion that wing flaps may require more vertical space when a hen is stationary (I assume as opposed to when locomotion as per L72). Or perhaps are you comparing the 49.5 cm in L69 to something else (I just do not see against what)? I think there is a piece of explanation missing that needs to be made explicit to a reader who is less familiar with the topic to make sure everyone can follow your train of thought.

L81-82: Have another look at this sentence, the flow felt a little of with the ‘to explore’ and ‘to measure’. This could be phrased more clearly.

Methods:

See general comments re: measurements

L97-98: hens were housed individually in cages that did not allow wing flapping based on the dimensions (42 cm height) and the results from this study. This may actually be a nice discussion point ( though ethically less nice for the birds in the individual cages). Curiosity question, but is that standard practice? It sounds like they were there from when they left the pullet phase so that would have been nearly 20 weeks…that’s definitely a long restriction.

L100: I’d suggest to reshuffle this paragraph a bit to go in chronological order – so starting with describing the pullet phase. Then there housing in the fertile egg flock. And then the mention that they were tested at 45 weeks of age. The chronological order will give people an easier time to follow, but also I have to admit that the way I first read it was ‘ok tested at 45 weeks old and then housed in cages of x dimensions’ and I wondered really hard how you would get hens to wing flap in those tiny cages! Obviously, that was not the intention but highlights how easy it is to lose your readers.

L108: I am picturing a ‘hard’ ruler, so not something that would bend and follow the contours of the body. Is that the right interpretation? What was the precision? Based on the data file it seems that some measurements may have been going in 0.5 cm increments.

L127-128: I assume you knew the height of the 5 gallon bucket and then checked if the right depth was reported? The pen was quite wide – how does the camera deal with the bird potentially not being in the centre but being or jumping more sideways of the camera (so at an angle) or did it not matter?

L166: where there any criteria for what qualified as a successful wing flap event? I am wondering if there were any case where there was for example a half-hearted one flap vs the more extensive flapping you can see in hens where they also really stretch their body upwards vs birds actually jumping up in an attempt to fly up. I assume the jumping up/fly option was excluded but better to be explicit. As for birds also stretching their body, could there be cases where their comb actually is the highest point over the wing itself?

L182-199 and Figure 2: I liked having the figure to help me visualize, but still have some question on how to read it. Could the figure images in panel A be numbered or perhaps put in order of processing steps? I feel the top right and lower left are clear and explained – the first is just the regular view, then you created the binary mask in yellow. Did an observer put the red dot on the higher vertical point or is that the program? It may be my eyes but I actually found it a bit hard to spot the red dot initially, so maybe that can be made clearer (and maybe it is also worth it to add it on the side view). But I do not know what I am looking at for the top left and the right bottom images.

I am also missing a small step describing on what is done next. You have the depth from the camera and that is then transformed into the maximum vertical height. So would that be e.g., the difference of the depth at the red dot say that is a depth of 200 cm and then you have the threshold of 245 cm so the maximum vertical height would of the wing flap would be 45 cm (or 50 cm if we include the 5 cm to the bottom of the floor?). I’m trying to make sure that I am really clear on the results of the ‘maximum vertical height’ being 51 cm (L216) is indeed the highest point of the wing from the floor. A small explanation here will be helpful to make it really clear and less room for ambiguity/interpretation errors.

L209-210: this reads as if the Shapiro-Wilk test was used for the correlation analyses while I assume the authors meant that the Shapiro-Wilk test was used to assess normality. How did you test for linearity?

Results:

I appreciate the results on the process itself, because I indeed would have had questions on how long it took for the authors to see the wing flapping movement or how many did or did not show the behaviour. Always nice when these sort of descriptive results are mentioned even if it is not the main purpose of the study.

I assume the two retested hens also showed wing flapping on the second attempt?

Table 1 (and associated text) – I understand these to be descriptive results in which case in my opinion it would be better to present the SD over the SE. You could also think of adding more parameters to get more insights into the distribution (e.g., median, Q1, Q3, CV) or consider presenting it as figures. Especially because it is a very descriptive study I feel there may be something more / value to be gained from your data.

Table 2 (and associated text) – based on the main comments this may change depending on if authors will make selections in what to include or not, but going from what is currently there I have a similar comment as for Table 1. I feel especially with the low numbers of animals with a relatively narrow range/variation of some variables (e.g., BW, folded wing) that it would be hard to find correlations in general (probably they are masked). So I think you would probably not find very strong correlations but the exercise/check is worthwhile. That said, I think that just showing the correlation itself and the P-value is not as insightful as, for example, scatter plots with the actual data points as that may also give ideas on potential outliers etc. If authors decide not to include this in the paper itself, I would recommend to consider submitting it as supplementary material.

Discussion:

Please consider all other comments and that any changes made should be reflected throughout the paper to make a cohesive story. This mostly comes together in the discussion. I have some more minor comments added below.

L238: ‘<50 cm’, I understand the ease but check throughout the discussion because I feel at times there is reference made to the 50 cm and other times 51 cm. Probably want to be consistent.

L252-271: I was expecting a larger focus on height as your main outcome variable, however, it only comes into play at around L266. I would rework this section a bit to highlight it more clearly and link it back to the rationale for why these correlations are important!

L255/L262: do you have an indication of what kind of numbers would be needed to confirm/discard the correlations?

L268: are you talking here about the cage dimensions in their original housing or the test set-up?

L281-282: ‘cage-reared and housed and was done as’ – I think some edits are needed here

L283: if Riddle et al. focused on when the flap occupied the most horizontal area does this then mean that Mench and Blatchford did not? That is sort of how it comes across to me right now.

L272-287: while this discussion is interesting, it is mainly focused on horizontal space and the two other studies. I miss how this is relevant or links to your current study and results. Of course, if the authors are able to add horizontal space then it becomes very relevant! Otherwise, right now it seems a bit excessive and repetitive from the introduction.

L297-298: I see you answered my earlier question here!

L298: ‘On one hand’ is this the right wording to use here? I’m expecting the on one hand – on the other hand statements to be contradictory and to me that is less the case here. For the second statement I am also a little confused because if Riddle never measured vertical space what did you compare to state that the hens in your study never occupied as much space while wing flapping as in their study (L304-306)?

If it was a mistake and it is supposed to refer to Mench and Blatchford then I also have questions on whether you can say ‘never’, because is 49.5 ± 1.8 cm really that different from your 51.0 ± 4.7 cm?

L311 is a heading of ‘additional considerations’ a journal requirement? It feels a little off, and I would rather have it worked into the regular discussion as a direction for future research/remaining questions. Probably it can be shortened as well as this was not the main focus of this study (see also earlier comment).

Conclusion

I actually miss here the conclusion regarding the actual value for the height; the authors did it nicely at the start of the discussion and I would encourage to show it here. I think that will help in case some readers only skim the conclusion to get the full picture.

L326-329: I would condense this a bit as the depth camera was more a tool to collect your data. I would not consider it a main conclusion but more a nice discussion point. I’ll leave it to the authors, it is likely a more personal preference and I will not be nit-picky about this.

L330: ‘provided interesting perspective’- check phrasing

L331: see earlier comment on the focus on cage-free systems – check if any adaptations are needed here.

L339: why especially strains with greater muscle strength or flapping abilities?

Other

Please double check the data file, I’d strongly recommend adding a legend fully explaining all headers and metadata to ensure others could potentially use the data. Some aspects I would consider would be for example,

- assuming FileName is actually the hen number?

- L wing meaning the left wing?

- Strain, age could be added

**Reviewer #3:**  This study examines the amount of vertical space required for hens to fully extend their wings while flapping from a stationary position. This study provides important information for properly constructing cage-free housing systems. The manuscript is clear and well written (see minor detailed comments below).

L35: change 'allow hens perform' to 'allow hens to perform'

L206: consider changing 'subjected to' to 'analyzed with'

L216: add average (or mean) before maximum height.

L248-251: this sentence needs rewording

**Reviewer #4:**  This manuscript describes the results of study, which used a depth camera to quantify the amount of vertical space needed for a hen to perform the wing flapping behavior. This outcome variable was correlated with body weight, and measurements of wing size. The topic is an important one. Many cage-free guidelines specify that hens must be able to wing flap without touching their enclosures, however the amount of space that is needed to satisfy this requirement has not been well established.

Addition of several critical methodological details will further enhance this manuscript. Specifically, 1) it is not clear how “wing flapping” was defined, or which set of wing movements were included, 2) it is not clear how the wing lengths were determined (where on the side of the body did the measurements originate), and 3) whether results were corrected for multiple comparisons on the same data. In addition to clarifying these key issues, I encourage the authors to consider that recording only the first flapping event could have biased the results, further limiting the external validity of the results. Additional feedback is provided below:

LN 15: Do “physical measures” refer to measures collected on the bird, or from the environment, or both?

LN 31: Consider rewording the phrase “Pushes to house hens” for readability.

LN 35: Change “perform” to “to perform”

LN 39-42: The connection between cage-free state regulations and their effects on imports should be clarified. As written, is not obvious that some of the state conventional cage housing bans (though not all) also mandate that eggs sold should come from cage-free housing, thereby influencing not only production, but also imports. Additionally, the sentence in LN 40 starts out discussing state bans, but ends up referring to corporation commitments. These impact imports in different ways.

LN 75: Should this read “including behavior of poultry” or similar?

LN 111: It is not clear where the measurement was taken for the “length of the physical structures of muscle and bone”. It seems that the measurement was taken to the distal tip of the longest phalange; where did it originate? A schematic would be useful.

LN 114-116: It is not clear where the measurement originated.

LN 133: I imagine that the second camera was not only capturing the side view, but potentially the front or back view, depending on how the bird oriented.

LN 134: Recording speed can affect the accuracy of measurements, especially when the target moves quickly. What were the recording speeds of the two cameras?

LN 153: When did habituation take place? Please describe the habituation procedure. Where the birds habituating to spending time in the crates, the test pen, or both?

LN 159: Should this read “space restrict”?

LN 162: Does this mean that the 2nd individual turned on the 2 cameras, or were they recording something else (e.g. the number of wing flaps)? If they turned on the cameras, did they do so manually, or were they connected to a computer, or trigger system?

LN 166: What constituted a “wing flap”? Was only the 1st movement, or set of movements, recorded? Could this have biased the estimates? For example, is it possible that the movements get larger the more a bird moves her wings? Were there any criteria as to whether the birds had to maintain their feet on the ground vs. if the birds were moving or jumping (e.g. if wing flapping while startled)? The position of the animal in vertical space would influence the readings.

LN 189-190: What does “The binary mask was then applied to the aligned RGB frame for visual 190 verification that no ‘hen pixels’ had been missed.” Mean? Is a binary mask a black and white image?

LN 195: This section seems to describe processing of the depth camera files, which resulted in an video of the bird with a red outline moving around it delineating the furthers point of the wing away from the body in each frame. The statistical analysis seems to use a single “vertical hight” value per bird (no repeated measures are used). How was this value extracted from the processed video?

LN 205-211: The measures referenced in the statistical section seem to refer to the measures taken on the hen. How was the video data (vertical distance of the wing flapping motion) statistically analyzed? It is not clear what type of “correlation analysis” as used: was a correlation matrix used, or were the individual comparisons shown in Table 1 ran as separate analyses. If the latter is the case, were the results corrected for multiple analyses?

LN 243-247: The size of the cages could be re-emphasized to support the statement regarding limited mobility due to housing.

LN 253: “Marginally significant” seems to imply that significance was reached, which is not the case.

LN 301-302: It is not clear why the behavior is classified as a “rebound” behavior. It is not obvious why one would classify it this way solely based on the amount of time it took for a hen to wing flap. There is no record of observations being conducted on the birds in their home pens or in the carriers. It is, therefore possible that they engaged in some form of wing flapping prior to being placed in the test pen. The classification of the behavior as a “rebound” behavior needs to be justified. Additionally,the reference to “stretching” in this sentence is confusing as wing stretching vs. wing flapping are different behaviors and may have different functions, as detailed in the introduction.

LN 309: Are you implying that these movements were the result of uncoordinated wing flapping? Since hens wing flap to regain balance, couldn’t it be the other way around?

LN 332: The birds’ perception was not assessed, and it is not clear how the measured outcomes relate to what a hen may perceive in terms of her needs.

LN 334-335: This implies that hens are equally likely to wing flap in all areas of the cage-free system. Is it important that hens be able to wing flap in all spaces, or is it sufficient if they can do so while on the floor of the aviary?

**Reviewer #5: ** This manuscript addresses an important issue of space requirements for wing flapping, particularly in cage-free housing that is an increasingly common housing type for laying hens. The authors utilized depth perception imaging to determine the height of wing flapping in Hy-Line W36 hens and correlated this to various physical characteristics. This is interesting, fundamental science to improve our understanding of wing flapping in farmed chickens. Broadly, I have concerns about the overall construct validity of this work since the researchers used caged hens instead of cage-free hens. The authors have already acknowledged and addressed this issue in thoughtful ways in the Discussion. As a result, further elaboration is not needed by the authors, but I wanted to acknowledge this concern as a reviewer as this study population choice constrains the validity of the results for cage-free hens, which was established as the rationale for the study in the introduction.

For comments for the authors to address, I have mostly minor comments about wording, missing information, and grammar (e.g., run-on sentences, improper commas) that made the manuscript difficult to follow at times. Please consider the following:

Introduction: Context about the variable height of cage-free aviaries (e.g., single-tier up to three- and even four-tier aviaries) would provide helpful, relevant rationale for evaluating vertical height needed for wing flapping as hens traverse the heights of structures, or at least heights between tiers to align with the later context described for wing flapping in the Discussion (L238-240) and Conclusion (L332-335).

L66-67: These numbers were cited in the discussion on L275-277. I originally interpreted the introduction as implying that one number was for white hens and one was for brown due to the ending phrase “brown feathered hens requiring less space than white”. However in the discussion, it is implied that both numbers are for Hy-Line W36 white hens. After double checking the references, the numbers do appear to correspond only to Hy-Line W36 hens. Please clarify the introduction phrasing to indicate that the numbers are only for Hy-Line W36 white hens.

L204-211: The statistics section is entirely missing information about the depth/height of wing flapping metric. Although the methods for generating these values are described above, correlating height of wing flapping with physical measurements is missing from the statistics section. Also, there should be a note about calculating descriptive statistics for Table 1.

L303-307: I am struggling to follow this logic, but I think I see where you are going. When you state that birds never occupied as much space as birds in Riddle, are you referring to vertical or horizontal space? It appears that you only measured vertical space in your study, but the other numbers cited from Riddle in the introduction and discussion are horizontal space. Please clarify which space you are referring to, and it will improve the logic of this section of the paragraph. Removing “On the other hand” in L304 would also help to improve clarity. If you are referring to horizontal space, then more details and results would need to be presented about how you measured horizontal space and what you found since you have only presented vertical space outcomes.

Data Excel File: Hen ID is missing from the “Depth Camera Height” tab, which would be critical for readers wishing to re-run the correlation analyses between height and physical measurements of hens. Also on this tab, it states on L 32 that pixel height was converted to mm before being exported, but all data are presented in either cm or inches. Is L32 supposed to be “converted to cm” instead? On the “Physical Measurements” tab, Column A, Line 32 is labeled as “DS”, which I assume is supposed to be “SD” for Standard Deviation given the formula in the corresponding row. The manuscript reports SEM in Table 1, so this calculation should also be included in the Excel file on both tabs for each metric to align with Table 1 values.

Grammatical errors

L29: There should be a semicolon between ‘region’ and ‘in’, so that it reads: “… needs of a region; in others …”

L34: Remove the comma between ‘laws’ and ‘are’, so that it reads: “… or laws are decreased stocking…”

L44: There should be a comma between ‘enclosure’ and ‘and’, so that it reads: “… their enclosure, and eggs produced…”

L72: There should be a comma between ‘stationary’ and ‘such’, so that it reads: “hen is stationary, such as”

L78: Cage free is not hyphenated here, but it is hyphenated in all other areas of the manuscript.

L80: There should be a comma between ‘legislation’ and ‘as’, so that it reads: “… enforcing legislation, as well as…”

L115: There should be a comma between ‘wing’ and ‘and’, so that it reads: “… the wing, and the wing was gently…”

L228-229: Is the word extended supposed to appear twice? Difficult to follow. “…length of her extended wing extended to the tip…”

L274: There should be a comma between ‘hens’ and ‘which’, so that it reads: “…cage-free hens, which is critical…”

L276: No comma is needed after [24]

L281: No comma is needed after [18]

L320: There should be a comma between ‘place’ and ‘or’, so that it reads: “…a particular place, or she may not…”

6. PLOS authors have the option to publish the peer review history of their article (what does this mean? ). If published, this will include your full peer review and any attached files.

**Do you want your identity to be public for this peer review?** For information about this choice, including consent withdrawal, please see our Privacy Policy .

Reviewer #1: No

Reviewer #2: No

Reviewer #3: **Yes: ** Richard A. BLatchford

Reviewer #4: No

Reviewer #5: No

---

## [Author Response · Author response to Decision Letter 1]

16 Jan 2025

Reviewer #1: Comments to the manuscript PONE-D-24-45495

The "Introduction" chapter is well written and contains enough information to support the validity of the study.

AU: Thank you

However, I miss a clear hypothesis. The authors constructed the purpose of the study and in principle one can infer from the text what their hypothesis is, but it is not directly stated. In my opinion, the authors want to confirm or rule out that the use of depth cameras will be a better tool for measuring the vertical height of the aviary needed for wing flapping than absolute measurements of the body dimensions of laying hens.

AU: We have rewritten the first more clearly as an objective regarding how a depth camera can be used to capture information and the second as a scientific hypothesis about the correlation between height while wing flapping and other measurements from hens (lines 121-123).

L95: Does this mean it was a parental (reproductive) flock?

AU: No, the flock is used to generate fertile eggs that are sold to medical researchers. We have tried to clarify this—and that the purpose of the hens was the reason they were caged (lines 136-137).

L95: Why was it decided to use Hy-Line hens despite the fact that the Authors indicated that similar studies had already been conducted for this genetic group (L69) and the influence of the hens' genotype on the aviary/cage space requirement had been confirmed (L67)? Wouldn't it be better to choose another laying hens' genotype that had not been studied in this respect?

AU: We used Hy-Line W36 again as we wanted to verify that the height recorded by Mench and Blatchford (2014) when these hens wing flapped during a jump was similar to a wing flap performed by a stationary hen with her feet on the ground who was not wing flapping for locomotion but likely was doing so for comfort behavior. We deliberately wanted to make this comparison before moving to other strains.

L97-98: I am not sure whether keeping hens individually in cages for the experiment was the right choice. Yes, it will give an answer about how much space a single hen needs, but as the Authors themselves mentioned in the introduction, it is important to indicate whether all the hens in the cage will be able to flap their wings simultaneously (L47-49). Considering that in commercial keeping of laying hens they are never kept individually, a group assessment seems more important!

AU: We measured each hen individually in a test pen and would have done so for the purpose of this experiment regardless of the hens’ home housing as our purpose was to see the maximum vertical height used by a hen that was unconstrained by her physical surroundings or crowded by other hens. We are assessing group synchrony, social facilitation and circadian rhythm of wing flapping in groups of hens in a later project.

L104-117: Why were only the wing dimensions measured? In my opinion, the entire body of the hen should be measured, i.e. from the foot to the end of the longest flight feather (when the hen is standing). The wing span (both) provides information on the width of the cage required, but for assessing the height required, the span of the hen's body from the foot to the end of the wing stretched upwards is important. Furthermore, I do not know whether body weight provides significant data in this case because it may not be related to body dimensions, e.g. hens with more fat and those with an egg in the shell gland will be heavier than others, even though their body dimensions may be smaller.

AU: We would love such a measure. However, it is almost impossible to ‘ask’ a hen to stand on the ground and then to stretch her wings up and extend them fully. If she does this she is flapping (not still for a measure). If a human tries to stretch her into this position while taking a measure, she will resist and try to pull her wings back in and could injure herself. She would also not be showing a normal standing posture.

We wanted to verify that there was not a correlation between weight and vertical height for the very reasons you mentioned. As no one had definitively shown this before, we thought it best to objectively measure it rather than assuming. We have added some text to explain this further in lines 90-120.

Reviewer #2: General

This paper addresses an interesting issue in space requirements, particularly vertical space, for lying hens while flapping their wings. There are two objectives to this study: 1) to assess the vertical space needed for hens to wing flap, and 2) to correlate BW and wing dimensions to maximum wing flapping height. For this purpose, the authors observed 28 Hy-Line W36 hens at 45 weeks of age, measured wing dimensions, and recorded them performing wing flapping behaviour with a depth camera.

The study is perhaps a little bit premature as it focussed on one strain and age; it would have been interesting to expand and e.g., assess vertical space over multiple strains and time points. That said, the study is well-described and a necessary first step. The authors also acknowledge the limitations and take care not to overstate their results. The paper is generally easy to read and follow, though I do have some points that I think should be clarified.

I have provided some of my main questions and concerns here, while more minor comments are described later.

1. Figure 1 and 2 do a really nice job of visualizing the experimental set up for the reader, but some images/schematics (e.g., a photograph with overlaid measurement indicators) to really show what is being measured with the ruler (and camera) would be a valuable addition. Reading some of the descriptions e.g., ‘length from carpal joint to the distal tip of the longest primary feather’ (L109), ‘where the muscle and bone ended at the distal tip of the longest phalange’ (L113), ‘measurements to the distal tip of the longest primary feather’ (L115), I lost oversight on what for example the differences was between what is being described in L113 and L115. Currently, it was not fully clear me e.g., where do you start/end the measurements, did you measure to a specific primary feather, a straight line from A to B or some other angle/method e.g., 90 degree from body centre etc.? It would help put the described measures into perspective and also make it easier to link/interpret the results on the correlations in Table 2.

AU: Yes, our initial description of the wing measures was confusing and incomplete. We have rewritten the text to clarify how we took each measure (lines 148-158) and added a new figure (Fig 1, caption text lines 161-169) to support the explanation.

2. I missed the rationale for the second objective (correlations between BW/wing dimensions and vertical height) which I think should be woven into the introduction / paper to help strengthen the story. Currently, it felt a little like it was there for the sake of having done some analyses. Some explanation on why you wanted to look at e.g., correlations between BW and wing dimensions, would be useful also to justify the multiple correlations evaluated (and the risk of finding correlations by chance). The ones between maximum height and the other variables are more intuitive to me but would also benefit if the authors could make this more clear and explicit throughout the paper.

AU: Thank you for your suggestion. We opted for the comparisons between the different measurements collected in order to provide an overview of the possible relations between birds’ body measurements. To the best of our knowledge, no previous studies investigated such relations in laying hens and this research represents the starting point for further explorations. Finally, although the multiple comparison could lead to an increase in type I errors, no significant correlation has been found in the present research, for which reason a correction for the multiple comparison was not applied. We added some explanations in the text (lines 289-290, 344-346).

3. The authors chose to focus on vertical space but I think it would have been really useful if the authors could also elaborate on the horizontal space. It might add useful insights if hens that need more vertical space also need more horizontal space as often assumed, but I have also heard of discussion where taller animals are thought to actually be slimmer/less dense and so taking up less space horizontally. If possible, I think this would really add value to the paper to also analyse horizontal space. If not possible, then I think the authors should carefully make their case for why it was not included.

AU: In hindsight, we wish we had done horizontal measures from these same birds. The findings of Mench and Blatchford reported horizonal space along with their vertical measurement, and the horizontal area was less than that observed by Riddle et al which focused on capturing maximum horizontal area alone. This suggests that your idea that less space would be occupied in the horizontal plane when the bird reaches maximum vertical height in the flap would be correct. These considerations have been discussed in the manuscript (lines 380-384).

Introduction:

There is a strong focus on the hens in cage-free housing throughout the introduction. However, would the space required not also apply to birds in caged housing system even if they have less opportunities to perform the behaviour?

AU: In theory, yes. In practice, hens in conventional cages likely have neither the horizontal or vertical space to flap at all. In large enriched/furnished cages, there may be horizontal space if the rest of the birds are clustered in a way that opens space near the hens, but cage heights are barely tall enough to allow birds to stand on short perches.

L41: should this not be export?

AU: This sentence has been edited.

L46-49: I think this is a fair question to raise, however its placing here made me think it would be addressed in the study. Perhaps nuance it a bit for some expectation management, or leave only a short mention the discussion?

AU: With respect, we prefer this as it is her but have strengthened the end of the introduction to clarify our objective and hypothesis.

L50: Wouldn’t the wings also require much more space in the horizontal place? You refer to this in L53

AU: True. We have removed the focus on vertical here (line 67) and just left it at more space generally before mentioning both horizontal and vertical in the next lines.

L51: a research group at the University of Guelph has done some studies in recent years on wing use/disuse in laying hens and its biological implications. In some they also reported on wing length and wing beat frequency, as well as musculoskeletal development. That may be useful to support the authors statements here, but more importantly give a bit more depth and up to date literature to the discussion especially around L243-247; L256-266 considering the angle taken there aligns well with these topics. See e.g., https://doi.org/10.1098/rsos.220809; https://doi.org/10.1098/rsos.230817; https://doi.org/10.1098/rsos.210196. I will leave it up to the authors to decide, but I wanted to point it out in case the authors were unaware.

AU: Thank you! We first focused our attention on the direct correlation between body measures and weight and related supportive references, thus we did not include the papers you indicated. However, we truly appreciate your perspective, and we believe that the inclusion of such manuscripts can provide a better overview of the reasons for which we did not see significant correlation, besides the limited number of birds available. We added these and some more references in the text (references 29-34; lines 110-120; 331-332; 352-353).

L62: I do not fully understand how the ‘while reducing plumage damage’ comes into play?

AU: Plumage damage was reported by EFSA as one of the indicators for identifying ideal space allowance together with behavioral evaluations. (If feathers repeatedly come into contact with cage wire or walls during performance of behaviors, they will become damaged.) That was not the main point we were trying to make, so the sentence has been rephrased to talk about animal-based indicators more broadly, which include behavior (lines 77-79).

L72-73: Could you elaborate on this? I am not sure how it leads to the suggestion that wing flaps may require more vertical space when a hen is stationary (I assume as opposed to when locomotion as per L72). Or perhaps are you comparing the 49.5 cm in L69 to something else (I just do not see against what)? I think there is a piece of explanation missing that needs to be made explicit to a reader who is less familiar with the topic to make sure everyone can follow your train of thought.

AU: the sentence has been modified and some more details added (lines 88-90).

L81-82: Have another look at this sentence, the flow felt a little of with the ‘to explore’ and ‘to measure’. This could be phrased more clearly.

AU: The sentence has been rephrased (lines 121-123).

Methods:

See general comments re: measurements

L97-98: hens were housed individually in cages that did not allow wing flapping based on the dimensions (42 cm height) and the results from this study. This may actually be a nice discussion point (though ethically less nice for the birds in the individual cages). Curiosity question, but is that standard practice? It sounds like they were there from when they left the pullet phase so that would have been nearly 20 weeks...that’s definitely a long restriction.

AU: Thank you for your suggestion. The housing system of the birds has been considered as factor of influence when discussing the results obtained (lines 329-330, 363-364, and 402-406). This housing is not standard practice for most hens at our farm. This is a special flock used for generating fertile eggs for biomedical research.

L100: I’d suggest to reshuffle this paragraph a bit to go in chronological order – so starting with describing the pullet phase. Then there housing in the fertile egg flock. And then the mention that they were tested at 45 weeks of age. The chronological order will give people an easier time to follow, but also I have to admit that the way I first read it was ‘ok tested at 45 weeks old and then housed in cages of x dimensions’ and I wondered really hard how you would get hens to wing flap in those tiny cages! Obviously, that was not the intention but highlights how easy it is to lose your readers.

AU: Thank you for the suggestion. The paragraph has been reorganized (lines 134-141).

L108: I am picturing a ‘hard’ ruler, so not something that would bend and follow the contours of the body. Is that the right interpretation? What was the precision? Based on the data file it seems that some measurements may have been going in 0.5 cm increments.

AU: It was a hard ruler and precision was 0.5cm increments. We have created a new figure showing how measurements were taken (Fig 1).

L127-128: I assume you knew the height of the 5 gallon bucket and then checked if the right depth was reported? The pen was quite wide – how does the camera deal with the bird potentially not being in the centre but being or jumping more sideways of the camera (so at an angle) or did it not matter?

AU: Yes, we knew the height of the bucket and checked it. Depth cameras are calibrated to deal with the subject of interest not being in the middle of the frame—otherwise they wouldn’t be much use. J

L166: where there any criteria for what qualified as a successful wing flap event? I am wondering if there were any case where there was for example a half-hearted one flap vs the more extensive flapping you can see in hens where they also really stretch their body upwards vs birds actually jumping up in an attempt to fly up. I assume the jumping up/fly option was excluded but better to be explicit. As for birds also stretching their body, could there be cases where their comb actually is the highest point over the wing itself?

AU: thank you for your comment, a part of the information was missing. The definition of successful wing flapping has been added (lines 209-216).

L182-199 and Figure 2: I liked having the figure to help me visualize, but still have some question on how to read

---

## [Decision Letter · Decision Letter 1]

20 Feb 2025

Maximum vertical height during wing flapping of laying hens captured with a depth camera

PONE-D-24-45495R1

Dear Dr. Siegford,

We’re pleased to inform you that your manuscript has been judged scientifically suitable for publication and will be formally accepted for publication once it meets all outstanding technical requirements.

Kind regards,

Arda Yildirim, Ph.D.

Academic Editor

PLOS ONE

Additional Editor Comments (optional):

Reviewers' comments:

Reviewer's Responses to Questions

**Comments to the Author**

1. If the authors have adequately addressed your comments raised in a previous round of review and you feel that this manuscript is now acceptable for publication, you may indicate that here to bypass the “Comments to the Author” section, enter your conflict of interest statement in the “Confidential to Editor” section, and submit your "Accept" recommendation.

Reviewer #1: All comments have been addressed

Reviewer #2: All comments have been addressed

Reviewer #4: All comments have been addressed

2. Is the manuscript technically sound, and do the data support the conclusions?

Reviewer #1: Yes

Reviewer #2: Yes

Reviewer #4: Yes

3. Has the statistical analysis been performed appropriately and rigorously? 

Reviewer #1: Yes

Reviewer #2: Yes

Reviewer #4: Yes

4. Have the authors made all data underlying the findings in their manuscript fully available?

Reviewer #1: Yes

Reviewer #2: Yes

Reviewer #4: Yes

5. Is the manuscript presented in an intelligible fashion and written in standard English?

Reviewer #1: Yes

Reviewer #2: Yes

Reviewer #4: Yes

6. Review Comments to the Author

Reviewer #1: (No Response)

Reviewer #2: The authors have clearly explained all my questions and concerns. The rationale and methods are much clearer now and limitations clearly discussed. I’ve only have some minor comments of more editorial nature but I do not need to sign off on those. Nicely done!

L84 ‘than white’ suggest to change to ‘than white feathered hens’

L103 ‘performance the behavior’ – ‘performance of the behaviour’

L177 Should FPS be written out fully here (note in L189 it is ‘frames per second’ fully written out)

L374 Double check [26] as I see [25] being the Riddle et al. paper and [20] being the Mench and Blatchford paper. [26] is currently listed in the reference list as Okinda et al.

Reviewer #4: The thoughtful revisions have further enhanced the quality of this manuscript. I have no further comments for the authors consideration.

7. PLOS authors have the option to publish the peer review history of their article (what does this mean? ). If published, this will include your full peer review and any attached files.

**Do you want your identity to be public for this peer review?** For information about this choice, including consent withdrawal, please see our Privacy Policy .

Reviewer #1: No

Reviewer #2: No

Reviewer #4: No

---

## [Editor Report · Acceptance letter]

PONE-D-24-45495R1

PLOS ONE

Dear Dr. Siegford,

I'm pleased to inform you that your manuscript has been deemed suitable for publication in PLOS ONE. Congratulations! Your manuscript is now being handed over to our production team.

Kind regards,

on behalf of

Prof. Dr. Arda Yildirim

Academic Editor

PLOS ONE